# *VAL* genes regulate vegetative phase change via miR156-dependent and independent mechanisms

Jim P. Fouracre[1], Jia He[1¤], Victoria J. Chen[1], Simone Sidoli[2], R. Scott Poethig[1]*

**1** Biology Department, University of Pennsylvania, Philadelphia, Pennsylvania, United States of America,
**2** Department of Biochemistry, Albert Einstein College of Medicine, Bronx, New York, United States of America

¤ Current address: Cold Spring Harbor Laboratory, Cold Spring Harbor, New York, United States of America
* spoethig@sas.upenn.edu

## Abstract

How organisms control when to transition between different stages of development is a key question in biology. In plants, epigenetic silencing by Polycomb repressive complex 1 (PRC1) and PRC2 plays a crucial role in promoting developmental transitions, including from juvenile-to-adult phases of vegetative growth. PRC1/2 are known to repress the master regulator of vegetative phase change, miR156, leading to the transition to adult growth, but how this process is regulated temporally is unknown. Here we investigate whether transcription factors in the *VIVIPAROUS/ABI3-LIKE* (*VAL*) gene family provide the temporal signal for the epigenetic repression of miR156. Exploiting a novel *val1* allele, we found that *VAL1* and *VAL2* redundantly regulate vegetative phase change by controlling the overall level, rather than temporal dynamics, of miR156 expression. Furthermore, we discovered that *VAL1* and *VAL2* also act independently of miR156 to control this important developmental transition. In combination, our results highlight the complexity of temporal regulation in plants.

## Author summary

During their life-cycles multicellular organisms progress through a series of different developmental phases. The correct timing of the transitions between these phases is essential to ensure that development occurs at an appropriate rate and in the right order. In plants, vegetative phase change—the switch from a juvenile to an adult stage of vegetative growth prior to the onset of reproductive development–is a widely conserved transition associated with a number of phenotypic changes. It is therefore an excellent model to investigate the regulation of developmental timing. The timing of vegetative phase change is determined by a decline in the expression of a regulatory microRNA–miRNA156. However, what controls the temporal decline in miR156 expression is a major unknown in the field. In this study we tested whether members of the *VAL* gene family, known to be important for coordinating plant developmental transitions, are critical regulators of

**Data Availability Statement:** The authors confirm that all data underlying the findings are fully available without restriction. The raw mass spectrometry data files generated in this study are

deposited on the freely accessible spectrometry data repository Chorus as part of Project No. 1706 (https://chorusproject.org/pages/dashboard.html#/projects/all/1706/experiments). The unprocessed and processed mass spectrometry data is also included in the S1 and S2 Datasets. The data underlying all other findings in this study are included in the S3 Dataset.

**Funding:** This project was funded by National Institutes of Health (www.nih.gov) Grant GM051893 to R.S.P. Research in the Sidoli lab is supported by the Umberto Mortari Award from Merck/MSD (2019), the Japan Agency for Medical Research and Development (AMED) and the New York Academy of Sciences (NYAS). The funders had no role in study design, data collection and analysis, decision to publish, or preparation of the manuscript.

**Competing interests:** The authors have declared that no competing interests exist.

vegetative phase change. Using a series of genetic and biochemical approaches we found that *VAL* genes are important determinants of the timing of vegetative phase change. However, we discovered that *VAL* genes function largely to control the overall level, rather than temporal expression pattern, of miR156.

## Introduction

Flowering plant development is underpinned by transitions between stereotypical stages of growth: embryogenesis, seed maturation, juvenile and adult phases of vegetative development and flowering [1]. The correct timing of these transitions is critical to plant survival and, ultimately, reproductive success. Vegetative phase change describes the transition from juvenile-to-adult vegetative growth and is associated with changes to multiple traits, including leaf morphology, light-use efficiency, herbivore resistance and shoot physiology [2–5]. In *Arabidopsis thaliana*, the juvenile phase is characterized by small round leaves that lack both trichomes on the abaxial surface and serrations. Adult leaves, on the other hand, are larger, more elongated, serrated and produce abaxial trichomes [6].

Vegetative phase change is triggered by activity of members of the *SQUAMOSA PROMOTER BINDING PROTEIN-LIKE* (*SPL*) family of transcription factors, which are post-transcriptionally repressed during juvenile development by the microRNAs miR156/miR157 [7–10]. miR156/miR157 are encoded by multiple genes of which *MIR156A* and *MIR156C* are the most functionally significant [11]. The expression of *MIR156A* and *MIR156C* declines during juvenile growth [12,13], leading to the de-repression of their *SPL* targets and the transition to adult growth. Elucidating what controls the decline in *MIR156A/C* expression is therefore critical to understanding how the juvenile-to-adult transition is regulated in plants.

The molecular mechanisms that lead to the temporal repression of *MIR156A/C* are only beginning to be understood. The activity of Polycomb Group (PcG) transcriptional repressors appears critical. There are two functional complexes of PcG proteins in plants, both of which repress gene expression through covalent histone modifications. PcG repressive complex 1 (PRC1) consists of a H2A E3 ubiquitin ligase module containing one AtBMI1 protein (AtBMI1A/B/C) and one AtRING1 protein (RING1A/B). PRC1 represses gene expression through ubiquitination of H2A (H2AK121ub) [14–16]. The PRC2 complex includes histone methyltransferases such as *CURLY LEAF* (*CLF*) and *SWINGER* (*SWN*) and promotes H3 trimethylation (H3K27me3) [17,18].

We have previously found that H3K27me3 increases at *MIR156A/C* in a PRC2-dependent manner during juvenile development, and that vegetative phase change is delayed in *swn* mutants [19]. The temporal deposition of H3K27me3 is accompanied by depletion of the antagonistic H3K27ac mark that is associated with active transcription. miR156 accumulation is also repressed by PRC1, as *atbmi1a/b* mutants exhibit delayed vegetative phase change [20]. In addition, we have found that accumulation of the active histone mark H3K4me3 decreases at *MIR156A/C* during vegetative development [21].

The findings that H3K27me3 replaces H3K27ac and H3K4me3 at *MIR156A/C* over time, and that PRC1/PRC2-activity promotes vegetative phase change, led us to propose that the temporal dynamics of miR156 accumulation are coordinated by antagonistic patterns of active (H3K27ac, H3K4me3) and repressive (H3K27me3) histone modifications [19,21]. In this model the stochastic removal of H3K27ac/H3K4me3 facilitates the deposition of H3K27me3 and the gradual epigenetic silencing of miR156. Similar mechanisms have been reported to function at other developmental transitions [22]. For example, during flowering, H3K27 deacetylation is a pre-requisite for PRC2-mediated H3K27me3 deposition at *FLOWERING*

*LOCUS C* (*FLC*) [23], and during seed maturation, PRC1 promotes the exchange of H3K4me3 for H3K27me3 at *DELAY OF GERMINATION1* (*DOG1*) and *ABSCISIC ACID INSENSITIVE3* (*ABI3*) [24].

Although there is good evidence that *MIR156A/C* are epigenetically silenced during vegetative development, how this mechanism is regulated temporally remains unknown. *VIVIPA-ROUS/ABI3-LIKE* (*VAL*) genes are excellent candidates for temporal effectors in this model. *VAL* genes encode B3 domain transcription factors that are closely related to the *ABI3/FUSCA3* (*FUS3*)/*LEAFY COYTLEDON2* (*LEC2*) clade of embryogenesis regulators. There are three *VAL* genes in *Arabidopsis*, of which *VAL1* and *VAL2* (also known as *HSI2* and *HSL2* respectively) are the most functionally important [25]. VAL proteins repress their targets by binding to 6 base pair RY-sequence motifs (CATGCA) via their B3 domain [26–32].

A number of observations suggest that *VAL* genes might provide the temporal information that coordinates vegetative phase change: 1) *VAL* genes regulate other developmental transitions, i.e. seed maturation [16,25] and flowering [30,32]; 2) *MIR156A/C* expression is elevated in *val1/2* mutants [20]; 3) VAL1/2 physically interact with several histone deacetylases (*HDA6/9/19*) [23,30,33,34]; and 4) *VAL* genes promote PRC1 and PRC2-binding [20,26,30,32,35].

In this study we investigated whether *VAL* genes function as temporal regulators of vegetative phase change. We report that reduced *VAL* activity significantly delays the timing of vegetative phase change through both miR156-dependent and independent mechanisms. We find that the temporal decline in miR156 expression is remarkably robust and is insensitive to loss of *VAL* function, inhibition of VAL1-binding and the combined loss of *VAL1* and PRC2 components. Finally, we show that the effects of *VAL1* on the timing of vegetative phase cannot be explained by temporal changes in its interactions with other proteins.

## Results

### *VAL* genes promote vegetative phase change

To investigate the role of *VAL* genes in vegetative phase change, we exploited a novel mutant we identified in an ethyl methanesulfonate screen for plants exhibiting prolonged juvenile development. Mapping-by-sequencing revealed a substitution at the *VAL1* locus, resulting in the conversion of a highly conserved arginine residue in the N-arm of the VAL1 B3 DNA-binding domain to a cysteine (S1A Fig). This arginine residue is critical for VAL1 binding to target RY-motifs [36]. The mutation in *VAL1* was confirmed to be the cause of the late juvenile phenotype by its failure to complement the null *val1-2* T-DNA insertion allele, and by the ability of the *VAL1* genomic sequence to rescue this phenotype (S1B and S1C Fig). Unlike *val1-2*, the novel *val1* allele is semi-dominant, and delays vegetative phase change when heterozygous (S1D Fig). We therefore named this new allele *val1-5(sd)*, consistent with the nomenclature of existing *val1* alleles [37].

Both *val1-5(sd)* and *val1-2* exhibit delayed vegetative phase change, with *val1-5(sd)* having a stronger effect on the timing of abaxial trichome production than *val1-2* (S1B Fig). As *VAL1* functions redundantly with *VAL2* to regulate other developmental transitions [25,30,32,38], we tested the effects of *val1; val2* double mutants on vegetative phase change. Previous analyses of *VAL* gene function have utilized *val1-2; val2-1* and *val1-2; val2-3* double mutants. However, seedling development is so strongly perturbed in *val1-2; val2-1* and *val1-2; val2-3* plants [16,25,35,38] that analyses of vegetative growth is problematic in these backgrounds. Therefore, we generated new *val1; val2* combinations using *val2-3* and a previously uncharacterized T-DNA insertion allele we named *val2-4*. The *val2-4* T-DNA is inserted in the last exon of *VAL2* (Fig 1A) and reduces *VAL2* transcript accumulation by about 60% (S1E Fig). Consistent with previous studies [25,32,38], *val2* single mutants had no discernible effect on vegetative

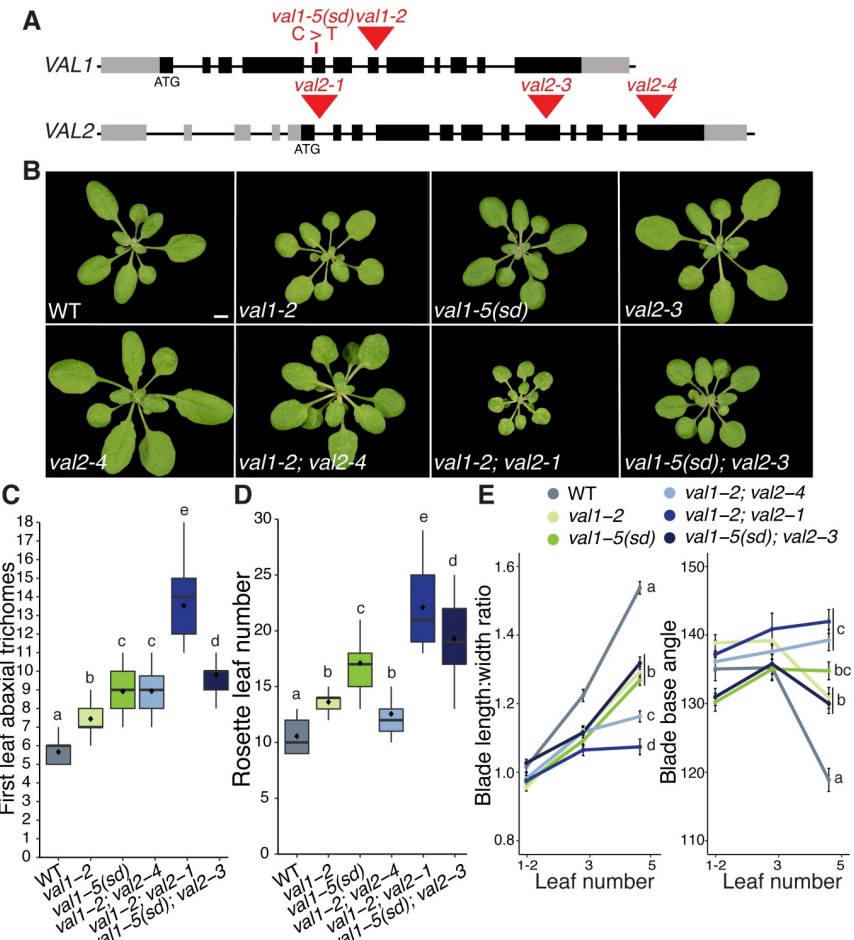

**Fig 1. *VAL* genes redundantly regulate vegetative phase change.** (A). Schematic of *val1* and *val2* alleles used in this study–grey boxes represent UTRs, black boxes represent exons, red triangles represent T-DNA insertions, red line represents EMS-induced base substitution. (B) Phenotypes at 21 DAG in LD conditions, scale bar = 5mm. (C-E) Quantitative analysis of vegetative development. Statistically distinct genotypes were identified by one-way ANOVA with *post hoc* Tukey multiple comparison test (letters indicate statistically distinct groups $P < 0.05$; for (E) comparisons were made at leaf 5), all plants grown in LD. (C,D) Boxes display the interquartile range (IQR) (boxes), median (lines) and values beyond 1.5*IQR (whiskers); mean values are marked by ◆. (E) Colored lines represent the mean and black bars the SEM. Sample sizes (C, D) 21–46, (E) 13–46.

phase change (Figs 1B and S1F). However, loss of *VAL2* activity enhanced the phenotypes of *val1-2* and *val1-5(sd)*. *val1-2; val2-4* and *val1-5(sd); val2-3* both exhibited delayed abaxial trichome production relative to *val1-2* and *val1-5(sd)*, respectively (Fig 1C). *val1-5(sd); val2-3* flowered significantly later than *val1-5(sd)* (Fig 1D) and *val1-2; val2-4* produced leaves that were significantly more juvenile in shape (i.e. rounder) than *val1-2* (Fig 1E). Neither double mutant combination was as phenotypically severe as *val1-2; val2-1* (Fig 1C, 1D and 1E). The weaker phenotype of *val1-5(sd); val2-3* compared to *val1-2; val2-3* [16,35], and stronger effects of *val2* in the *val1-2* than *val1-5(sd)* background (Fig 1E), suggests that the semi-dominant phenotype of *val1-5(sd)* is mediated by interaction with *VAL2*. Importantly, the rate of germination was higher in *val1-2; val2-4* and *val1-5(sd); val2-3* relative to existing *val1; val2* double mutants. *val1-2; val2-4* and *val1-5(sd); val2-3* thus provide a balance between phenotypic strength and experimental viability and are useful tools for investigating the role of *VAL* genes in developmental timing.

## *VAL* genes function predominantly as quantitative–rather than temporal–regulators of miR156 expression

Vegetative phase change results from a temporal decline in miR156 expression [8]. A previous analysis of *val1-2; val2-1* revealed elevated expression of *MIR156A/C* at a single time point [20]. To determine whether the delay in vegetative phase change we observed in *val* mutants is associated with a general increase in the level of miR156, or with a delay in the decline in this miRNA, we quantified miR156 expression in the shoot apex and in isolated leaf primordia at different times in shoot development. The primary transcripts of *MIR156A* and *MIR156C* were expressed at similar levels, and exhibited a similar temporal expression pattern, in wild type, *val1-5(sd)* (Fig 2A and 2B) and *val1-2* shoot apices and leaf primordia (S2 Fig). However, the abundance of the mature miR156 miRNA transcript was significantly higher in *val1-5(sd)* leaf primordia than in wild type (Fig 2B), and it was also marginally higher in *val1-5(sd)* shoot apices than in wild type (Fig 2A). The *val1-5(sd); val2-3* (Fig 2A and 2B) and *val1-2; val2-1* (S2 Fig) double mutants had stronger effects on *MIR156A* and *MIR156C* expression than the respective *val1* single mutants, suggesting that *VAL1* and *VAL2* function redundantly to repress *MIR156A* and *MIR156C* transcription. Mature miR156 was elevated throughout development in both the shoot apices (Fig 2A) and the leaf primordia (Fig 2B) of *val1-5(sd); val2-3* double mutants. Loss of *VAL* activity also produced a slight increase in miR157 levels at later

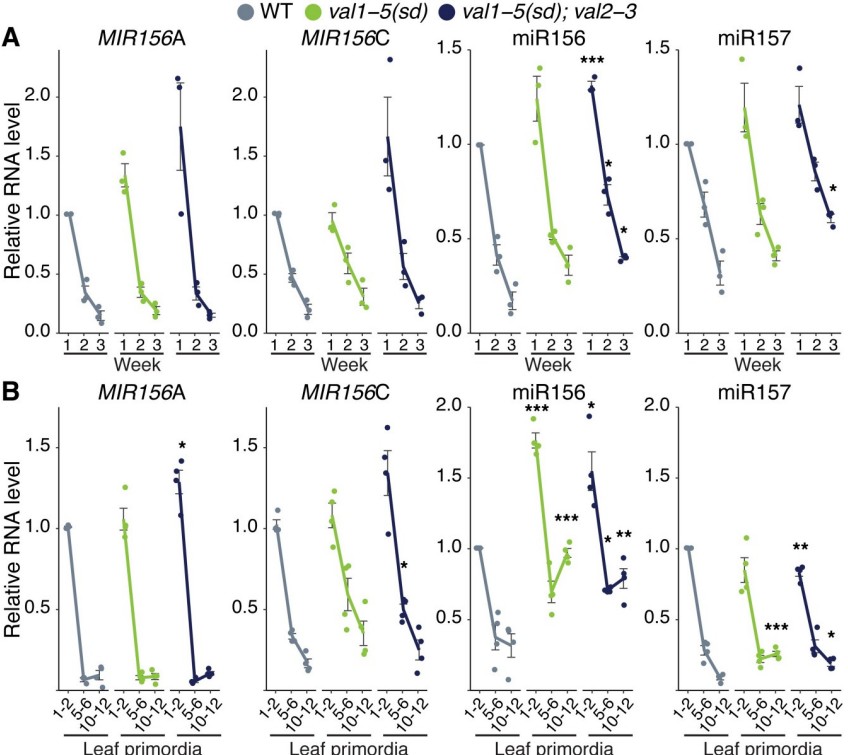

**Fig 2.** *VAL genes function predominantly as quantitative regulators of miR156 expression.* (A, B) qRT-PCR analyses of gene expression. (A) Shoot apices with leaf primordia (LP) ≥ 1mm removed at 1, 2 and 3 weeks. (B) Isolated LP 0.5-1mm in size. All plants were grown in SD conditions. Each data point represents a biological replicate and is the average of three technical replicates. Coloured lines represent the mean and black bars mean±s.e.m. Asterisks represent significant differences between WT and *val* mutants at the same time point, calculated by an unpaired two-tailed *t*-test with a Bonferroni correction for multiple comparisons (* $P < 0.025$; ** $P < 0.005$; *** $P < 0.0005$).

stages of development (Figs 2A and 2B and S2). Although *val1-5(sd); val2-3* increased the abundance of miR156/miR157, it had only a minor effect on the temporal expression patterns of these miRNAs. For example, miR156 expression decreased 2.36-fold between 1W and 2W and 2.31-fold between 2W and 3W in wild type plants, but decreased 1.77 and 1.82-fold between the same time points in *val1-5(sd); val2-3* plants (Fig 2A). Taken together, these data suggest that *VAL* genes function primarily as general, rather than temporal, regulators of miR156 expression.

### *VAL* genes coordinate PRC2 recruitment at specific *MIR156* loci

We have previously demonstrated that the temporal decline in *MIR156A* and *MIR156C* expression is associated with PRC2-dependent and progressive deposition of H3K27me3 at these loci [19]. To determine if *VAL* genes contribute to this process, we examined H3K27me3 accumulation in the *val1-5(sd); val2-3* double mutant. As previously reported [19], H3K27me3 levels increased at the *MIR156A* and *MIR156C* loci during vegetative development in wild type plants (Fig 3A and 3B). Although there was no difference in the temporal pattern of H3K27me3 deposition at *MIR156A* in *val1-5(sd); val2-3*, the rate of H3K27me3 deposition at *MIR156C* was significantly slower in this double mutant (Fig 3B). These results are consistent with a recent genome-wide study that revealed a decrease in H3K27me3 levels at *MIR156C*, but not *MIR156A*, in *val1-2; val2-3* plants [35]. As a control, we measured H3K27me3 deposition at the floral regulator *FLC*. In the absence of vernalization, we observed no change in the level of H3K27me3 during vegetative development in wild type plants (Fig 3B). However, consistent with previous reports [30,32], there was a significant decrease in H3K27me3 at *FLC* in *val1-5(sd); val2-3*.

*VAL1* is thought to act by recruiting PRC1 which, in turn, promotes the activity of PRC2 [15,39,40]. As a genetic test of this hypothesis, we examined the interaction between *val1-2* and *clf-29* and *swn-3*, loss-of-function mutations in the functionally redundant genes that encode the histone methyltransferase activity of PRC2. As we have shown previously [19], *swn-3* had a larger effect on the timing of vegetative phase change than *clf-29* (Fig 3C and 3D). Consistent with the hypothesis that *VAL1* regulates vegetative phase change via its effect on PRC2 activity, *clf-29* and *swn-3* interacted synergistically with *val1-2*, in that the double mutants had a much more severe vegetative phase change phenotype than the single mutants (Fig 3C and 3D). Notably, *val1-2* suppressed the curling leaf phenotype of *clf-29* (Fig 3C), presumably because it enhances *FLC* expression [41]. Together, these results suggest that *val* mutations delay vegetative phase change by interfering with the activity of PRC2.

To determine whether the synergistic interaction between *clf-29* and *swn-3* and *val1-2* is due to enhanced miR156/miR157 expression, we quantified expression of the mature miR156 and miR157 miRNAs, and the primary *MIR156A* and *MIR156C* transcripts, in these mutant backgrounds. The overall level and expression pattern of the mature miR156/miR157 transcripts, and the primary *MIR156A* and *MIR156C* transcripts, were not affected by *val1-2*, or by *val1-2; clf-29* and *val1-2; swn-3* double mutants (Fig 3E). These results suggest that *VAL* genes temporally regulate the deposition of H3K27me3 at specific *MIR156* loci, but are not necessary for the temporal decline in miR156 expression.

*VAL* genes also repress gene expression by promoting H2AK121ub deposition via recruitment of PRC1 [16,39]. Unlike H3K27me3 (Fig 3B) [19], we found no evidence that H2AK121ub increases consistently over time at *MIR156A* and *MIR156C* (Fig 3F). There appears to be a transient peak of H2AUb deposition at 2W of growth for *MIR156A*, *MIR156C* and *FLC*, however, this finding is supported by only 2 biological replicates. *val1-5(sd); val2-3* had lower levels of H2AK121ub than wild type plants at 2W but we observed no effect of loss of *VAL* activity at 1W or 3W.

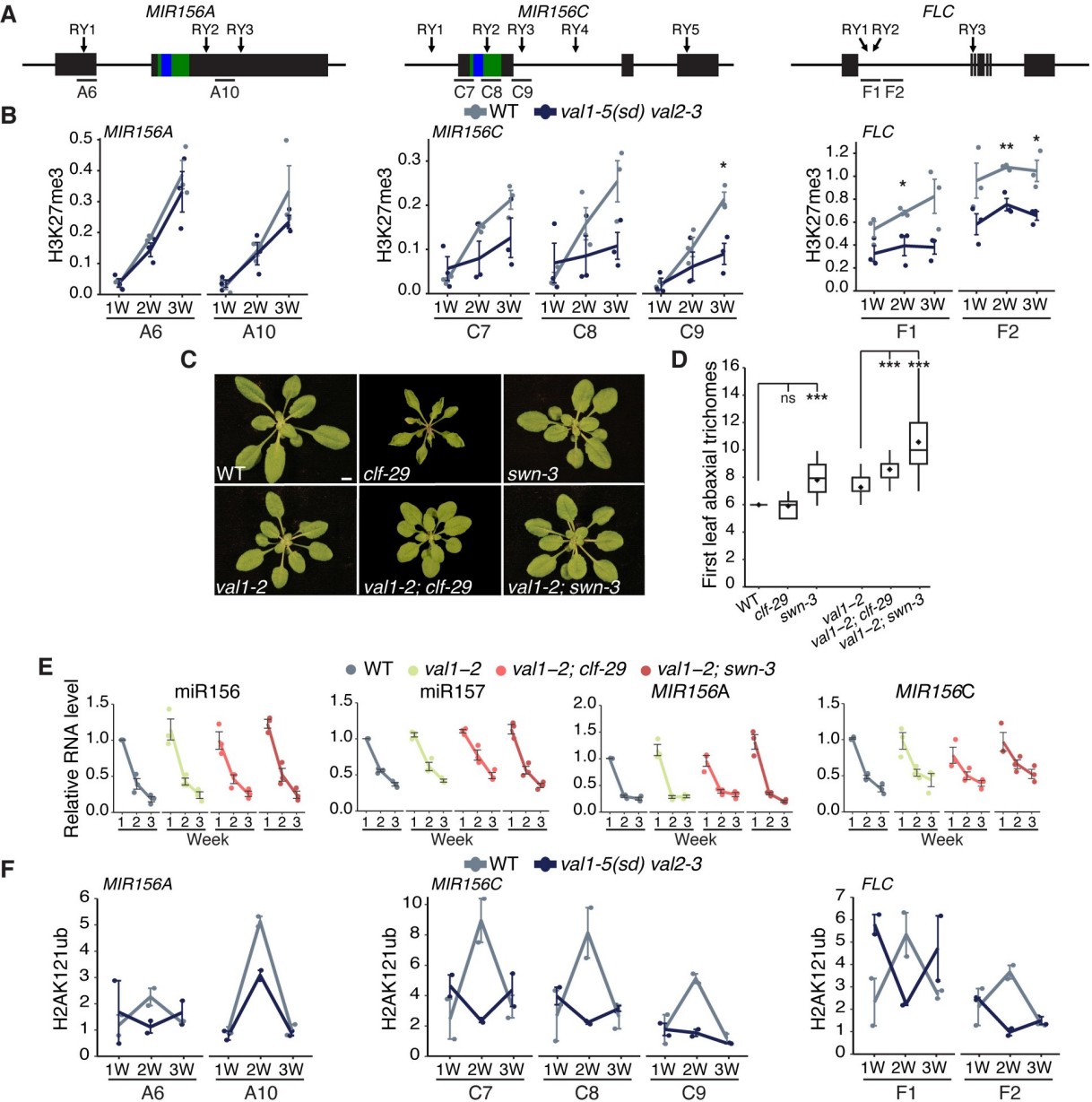

**Fig 3. *VAL* genes regulate miR156 activity via-chromatin modifications.** (A) Schematics of the primer locations used for ChIP-qPCR. Blue and green bars represent sequences encoding the mature miRNA and miRNA hairpin respectively. (B) Temporal analysis of H3K27me3 by ChIP-qPCR. Each data point represents a biological replicate and is the average of three technical replicates. Lines represent the mean, bars represent the mean±s.e.m., asterisks represent significant differences between WT and *val1-5(sd); val2-3* at the same time point, calculated by an unpaired two-tailed *t*-test (* *P* < 0.05; ** *P* < 0.01; *** *P* < 0.001). H3K27me3 values are relative to H3 and normalised to *STM* as an internal control. Plants were grown in SD conditions. (C, D) Phenotypes in LD. (C) Photographs taken at 21DAG, scale bar = 5mm. (D) Asterisks represent significant differences to either WT or *val1-2*, calculated by unpaired two-tailed *t*-test with Bonferroni correction for multiple comparisons (*** *P* < 0.0005). Sample size 22–46. (E) qRT-PCR analyses of gene expression in shoot apices with LP ≥ 1mm removed at 1, 2 and 3 weeks. Each data point represents a biological replicate and is the average of three technical replicates. Coloured lines represent the mean and black bars represent the mean±s.e.m. Plants were grown in SD conditions. (F) Temporal analysis of H2AK121ub by ChIP-qPCR, values are relative to input and normalised to *ABI3* as an internal control. See (B) for details (no statisical analyses are included in (F) due to limited replication).

## *VAL1* regulates vegetative phase change via miR156-dependent and miR156-independent mechanisms

To investigate whether the effects of *val1-5(sd); val2-3* on the chromatin state of *MIR156A/C* are due to a direct regulatory interaction, we carried out chromatin-immunoprecipitation qPCR using an HA-tagged version of VAL1 [30]. Confirming the results of a recent ChIP-seq study [35], we found VAL1-binding at specific locations within both the *MIR156A* and *MIR156C* loci (Figs 4A and 4B and S3). The affinity of VAL1 for *MIR156A/C* appeared consistent throughout vegetative development.

VAL1/2 bind to RY-sequence motifs, of which there are multiple copies in both *MIR156A* and *MIR156C* (Fig 4A). To determine if these RY-sites are required for the regulation of vegetative phase change, we mutated 5 RY-sites in *MIR156C* individually and in combination. We selected *MIR156C* because it is more sensitive to *VAL* activity than *MIR156A* (Figs 2,3 and S2). A C>T substitution that eliminates VAL1 binding [36] was introduced in one or all of these sites in a genomic construct of *MIR156C*. Wild type and mutant constructs were then transformed into a *mir156a; mir156c; mir157a; mir157c quadruple mutant* (*qm*) background. We chose this background because it has a low level of endogenous miR156/miR157 activity and is therefore sensitive to small changes in the level of miR156 [11]. Plants transformed with a wild type *MIR156C* construct (*+RY*) produced leaves with abaxial trichomes at the same node as the *mir156a; mir157a; mir157c triple mutant* (*tm*) (Fig 4C). *tm* has an endogenous copy of *MIR156C*, confirming that the transgenic *MIR156C* sequence is fully functional. Deletion of individual RY-sites produced a significant delay in the timing of abaxial trichome production relative to *MIR156C +RY*, and deletion of all 5 RY-sites produced a more significant delay than deletion of any single site (Fig 4C). A similar result was obtained in the case of the angle of the leaf base (Fig 4D). These results demonstrate that all five RY-sites are important for the expression of *MIR156C*, and that they function additively. Individual RY-sites have also been shown to interact additively to repress the VAL1-PRC2 targets *FLC* and *DOG1* [27,32].

To determine if these phenotypic effects are due to altered *MIR156C* expression, we quantified miR156 levels in *MIR156C +RY*, *MIR156C -RY2* and *MIR156C -RY12345* plants. *MIR156C -RY2* was selected because it has a marginally stronger effect than other individual -RY deletions (Fig 4C and 4D). Although there was considerable variation in miR156 levels between independent transgenic lines, *MIR156C -RY2* and *MIR156C -RY12345* plants had significantly more miR156 than plants transformed with *MIR156C +RY* (Figs 4E and S4). However, the temporal expression pattern of *MIR156C* was identical in *-RY* and *+RY* plants.

To establish whether the effects of RY-deletion are *VAL1*-dependent, we crossed *val1-2* into the *qm; MIR156C +/-RY* lines. If the delay in vegetative phase change in *MIR156C -RY* lines is a consequence of reduced VAL1 binding, *val1-2* should have less effect in *MIR156C -RY* lines than in *MIR156C +RY* lines or the *tm*, in which RY sites are intact. Surprisingly, we found that loss-of *VAL1* significantly delayed abaxial trichome production in *MIR156C -RY* as well as *MIR156C +RY* and *tm* plants (Fig 4F). It is possible that *MIR156C* RY-sites are bound by other B3 domain transcription factors. However, RY-binding is restricted to the *ABI3/FUS3/LEC2* and *VAL* clade of B3 domain genes [32,36,42], whose expression of is largely restricted to seed development [43]. It is therefore unlikely that these genes regulate vegetative shoot identity. Moreover, FUS3 and ABI3 directly promote the expression of *MIR156C* [44,45]. A role for these genes in the regulation of *MIR156C* post-germination is thus inconsistent with the juvenilized phenotype and elevated miR156 expression we found in *MIR156C -RY* plants. (Fig 4C, 4D and 4E). With regard to the potential effects of other *VAL* genes, we observed no vegetative phase change phenotype in *val2* single mutants (Figs 1B and S1F) and *VAL3* has limited expression and functionality relative to *VAL1* and *VAL2* [25]. Alternatively, this result suggests

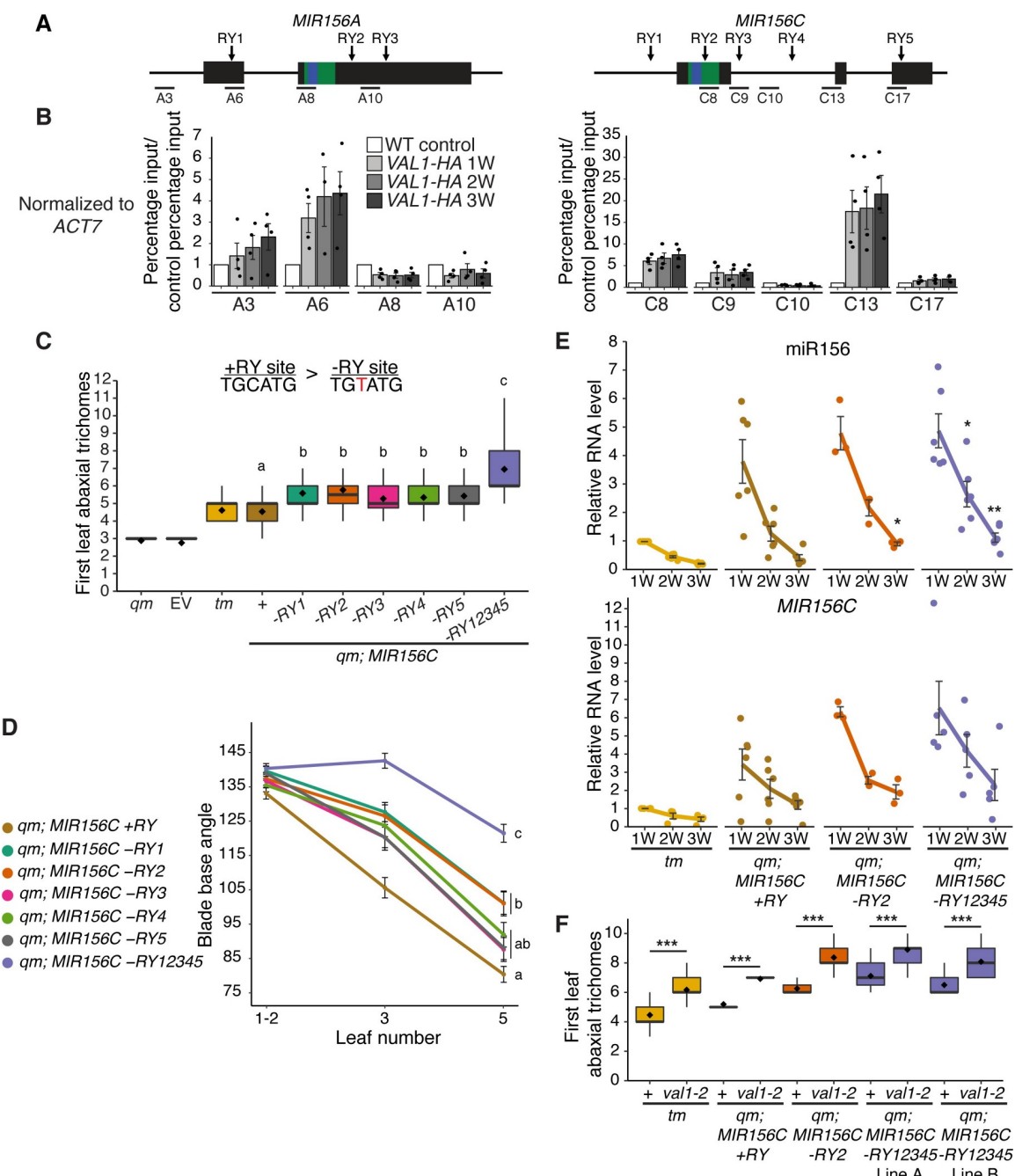

**Fig 4. Loss of RY VAL-binding motifs at the *MIR156C* locus delays vegetative phase change.** (A) Schematic depicting the location of primers used for ChIP-qPCR, the sequences encoding the miR156 hairpin and mature miRNA are coloured green and blue respectively. (B) Anti-HA ChIP-qPCR of WT Col control plants at 2W and *VAL1::VAL1-HA; val1-2; FRI-Sf2* plants at 1, 2 and 3W of growth. The data is presented as percentage input normalized to *ACT7* and is displayed relative to WT. Each data point represents a biological replicate and is the average of three technical replicates, bars represent the mean and error bars the mean±s.e.m. (C, D) Phenotypes of T1 plants transformed with *MIR156C* RY variants. Statistically distinct genotypes were identified by one-way ANOVA with *post hoc* Tukey multiple comparison test (letters indicate statistically distinct groups $P < 0.05$; comparison in (D) made at leaf 5). *qm = mir156a mir156c mir157a mir157c* quadruple mutant, *tm = mir156a mir157a mir157c* triple mutant, EV = empty vector. (D) Colored lines represent the mean and black bars the mean±s.e.m. Sample size (C) 26–52, (D) 37–51. (E) qRT-PCR analyses of gene expression in shoot apices with LP ≥ 1mm removed at 1, 2 and 3 weeks. Each data point represents an independent homozygous T3 line and is the average of three technical replicates. Colored lines represent the mean and black bars the mean±s.e.m. Asterisks represent significant differences between *qm; MIR156C* and *qm; MIR156C -RY* lines at the same time point, calculated by an unpaired two-tailed *t*-test with a Bonferroni correction for multiple comparisons (* $P < 0.025$, ** $P < 0.005$). (F) Genetic interaction between *val1-2* and *MIR156C* RY

deletions. Asterisks represent significant differences between plants with wild type or null *VAL1* alleles calculated by an unpaired two-tailed *t*-test (*** $P < 0.001$). Sample size 24–36. Phenotyping analyses were carried out in LD conditions, gene expression and ChIP analyses were carried out in SD conditions.

that *VAL1* may regulate vegetative phase change through both miR156-dependent and miR156-independent mechanisms. This interpretation is supported by the observation that loss-of *VAL1* and PRC2-components strongly delayed vegetative phase change but had only minor effects on miR156 expression (Fig 3C, 3D and 3E).

To test this hypothesis, we introgressed *val1-2* into the *qm* genetic background, which has low levels of miR156/miR157 [11]. Although the morphology (Fig 5A) of *val1-2; qm* leaves 1 and 2 were indistinguishable from that of *qm* plants, *val1-2* partially suppressed the effect of the *qm* genotype on the morphology of leaves 3 and 5 (Fig 5B). This confirms that *VAL1* functions through a miR156/miR157-independent mechanism to regulate vegetative phase change.

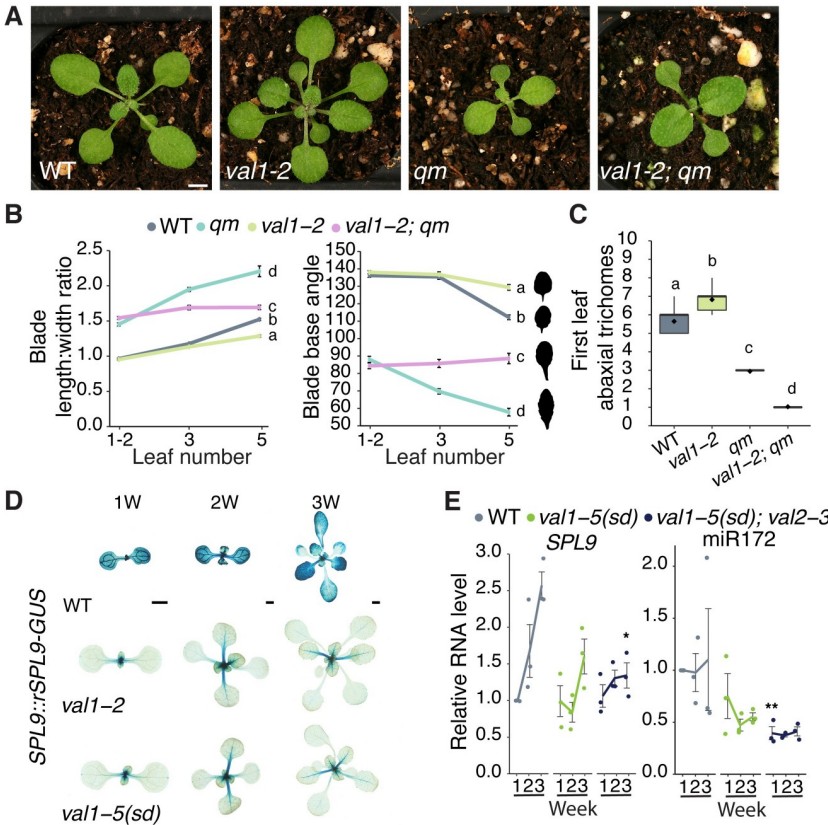

**Fig 5.** ***VAL1* regulates vegetative phase change by miR156-dependent and independent mechanisms.** (A-C) Phenotypes of *val1-2* and *mir156a mir156c mir157a mir157c quadruple mutant* lines. (A) Photographs taken at 17 DAG. Scale bar = 1mm. (B, C) Statistically distinct genotypes were identified by one-way ANOVA with *post hoc* Tukey multiple comparison test (letters indicate statistically distinct groups $P < 0.05$; (B) comparisons made at leaf 5). Bars represent the mean±s.e.m. Sample size (B) 17–60, (C) 18–36. Silhouettes in B show representative leaf 5 shapes. (D) Expression of a miR156-resistant (*rSPL9*) reporter construct in WT, *val1-2* and *val1-5(sd)* backgrounds. Scale bars = 1mm. (E) qRT-PCR analysis of gene expression in shoot apices with LP $\geq$ 1mm removed at 1, 2 and 3 weeks. Each data point represents a biological replicate and is the average of three technical replicates. Coloured lines represent the mean and gray bars represent the mean±s.e.m. Asterisk represents significant difference between WT and *val* mutant lines at the same time point, calculated by an unpaired two-tailed *t*-test with a Bonferroni correction (* $P < 0.025$; ** $P < 0.005$). All phenotypic analyses were carried out in LD conditions, the gene expression analysis was carried out in SD conditions.

Surprisingly, a survey of abaxial trichome production revealed that *val1-2* enhanced this aspect of the precocious *qm* phenotype (Fig 5C). This result can be explained by the observations that reduced histone deacetylation in a *toe* loss-of-function background accelerates abaxial trichome formation [46], that *SPL* genes repress *TOE* activity [9], and that *VAL* genes promote histone deacetylation [23,34,39].

To determine if *VAL1* regulates *SPL* gene expression independently of miR156, we crossed a miR156-resistant (*rSPL9*) *SPL9::rSPL9-GUS* reporter construct into *val1* genetic backgrounds. The expression of this reporter was visibly and strongly suppressed by *val1-2* and *val1-5(sd)* (Fig 5D), implying that miR156 is not required for the regulation of *SPL9* by *VAL1*. The transcript levels of *SPL9*, and its target miR172, were also decreased in *val* mutant plants (Fig 5E). However, it is difficult to know if this decrease is dependent or independent of miR156 because miR156 induces cleavage of the *SPL9* transcript [11,47,48].

VAL1/2 act as transcriptional repressors. As *SPL9* transcription decreases in *val* loss-of function mutants it is therefore unlikely that VAL1/2 regulate *SPL9* directly. To test this prediction, we quantified H2AK121ub and H3K27me3 at *SPL9*. Consistent with previous studies, we found high levels of H2AK121ub but negligible H3K27me3 at *SPL9* (S5 Fig) [40,49]. We also found no sustained difference in the abundance of these modifications in wild type and *val1-5(sd); val2-3* plants. Taken together these results suggest that, in addition to regulating *SPL9* via their effect on miR156 levels, VAL1/2 promote *SPL9* expression indirectly through one or more miR156-independent mechanisms (i.e. repression of an unknown transcriptional repressor of *SPL9*). The observation that *val1-5(sd); val2-3* has no effect on H2AK121ub at *SPL9* also suggests that *VAL1/2* are not universally required for PRC1-activity.

## The effects of *VAL1* on developmental timing may be partly explained by its expression pattern

Our results show that *VAL* genes control the timing of vegetative phase change (Fig 1C and 1E), and have subtle effects on the expression pattern of miR156 during vegetative development (Figs 2 and S2). To determine if these effects are attributable to changes in the expression level of *VAL1*, we measured the abundance of *VAL1* transcript levels during vegetative growth. We observed a small but significant increase in *VAL1* transcripts in shoot apices between 2 and 3 weeks of growth (Fig 6A) but there was no significant change in *VAL1* levels in leaf primordia (Fig 6B). To further investigate *VAL1* expression over time, we generated a *VAL1* transcriptional reporter by fusing a 2.3kb sequence containing the *VAL1* promoter and 5' UTR, and a 2kb sequence containing the *VAL1* 3'UTR and terminator, to the GUS coding sequence (*VAL1::GUS-VAL1 3' UTR)*. We generated a *VAL1* translational fusion by inserting a 3.8kb *VAL1* genomic sequence upstream of *GUS* in this construct (*VAL1::VAL1-GUS-VAL1 3'UTR*). Because the expression of the transcriptional fusion was consistently more diffuse, variable, and weaker than the expression of the translational fusion (S6 Fig), we used plants containing the translational fusion for subsequent studies.

During embryogenesis, the translational fusion was expressed in the root and shoot apical meristems and provasculature (Figs 6C and S6). Following germination, expression became restricted to the shoot and root apices and initiating lateral root primordia (Figs 6C and S6). Throughout the rest of shoot development, the translational fusion was expressed in the shoot apex and during the early stages of leaf development (Fig 6C). Histological inspection indicated that *VAL1* expression increases in the shoot apex during vegetative development. This was validated by a quantitative analysis of GUS expression, which demonstrated that VAL1 accumulates more strongly in the shoot apex than leaf primordia. Further, that VAL1 levels increase over time in the shoot apex but not in older leaf primordia (Fig 6E and 6F). Taken

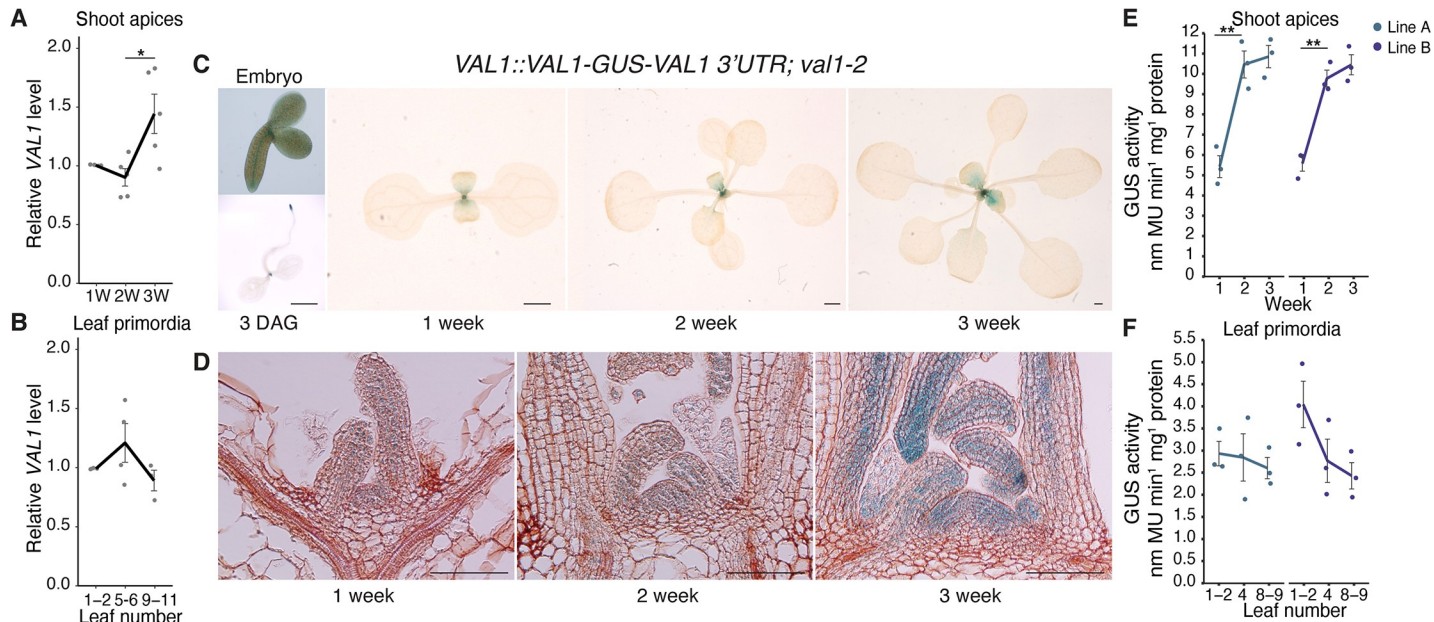

**Fig 6. *VAL1* is expressed throughout vegetative development.** (A, B) qRT-PCR analyses of gene expression. (A) Shoot apices with leaf primordia (LP) ≥ 1mm removed at 1, 2 and 3 weeks. (B) Isolated LP 0.5-1mm in size. (C-F) Analyses of a *VAL1-GUS* translational fusion. (C) Expression in whole plants, scale bars = 1mm. (D) Expression in shoot apices following wax sectioning, scale bars = 0.1mm. (E, F) VAL1 levels quantified by MUG assay in two independent homozygous T3 lines. (A, B, E, F) Each data point represents a biological replicate and is the average of three technical replicates. Lines represent the mean and grey bars mean±s.e.m. Asterisks represent significant differences between two continuous time points, calculated by an unpaired two-tailed *t*-test (* $P < 0.05$; ** $P < 0.01$). All plants except the 3 DAG sample (C–long days) were grown in SD conditions.

together, these data indicate that *VAL1* expression is restricted to apical meristems and the early stages of root and leaf development. Our data also indicate that *VAL1* expression increases during shoot development in very young leaf primordia, quickly declines to a uniform level as the leaf develops, and ceases before the leaf is fully expanded. However, the functional significance of increased VAL1 accumulation during vegetative development is unclear, as we did not detect a concomitant increase in VAL1-binding to *MIR156A/C* (Figs 4B and S3). Finally, the difference in the staining patterns and stability of our transcriptional and translational reporters suggest that *cis*-regulatory elements within the *VAL1* coding sequence regulate the level and site of its expression.

VAL1 has previously been found to physically interact with VAL2, multiple PRC1 and PRC2 components, and the transcriptional repressor SAP18 [16,26,27,29,30,32,33,50]. However, these studies do not provide information about VAL1's *in planta* protein interactions over time. To determine whether VAL1 interacts with different proteins at different stages of vegetative development, we carried out a mass spectrometry analysis of proteins bound to HA-tagged VAL1 at 1, 2 and 3 weeks of growth. Immunoprecipitations were carried out on a *VAL1-HA; val1-2; FRI-Sf2* line, using a *val1-2; fri* line as a control. The difference in the *FRI* genotype of these lines is a consequence of the genotypes available at the time the experiments were performed and may have had an effect on our results.

ATBMI1A was significantly enriched in the combined experimental samples relative to the control samples (Fig 7A), which is consistent with a previous mass spectrometry analysis of proteins bound to VAL1-HA [30]. We did not detect other proteins that have been identified by mass spectrometry in immunoprecipitation experiments with VAL1-HA. We also observed a highly significant enrichment of the chloroplast binding protein CRB. However, this is explained by the enrichment of CRB in the total proteome of the experimental versus control

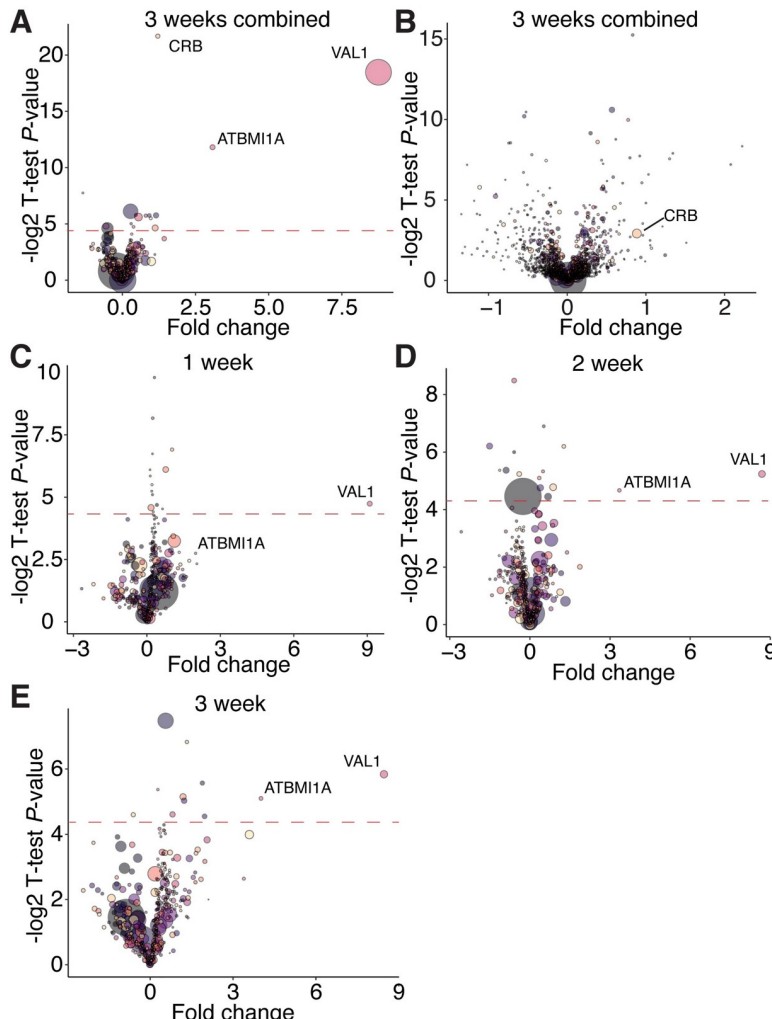

**Fig 7. VAL1-protein interactions are consistent during vegetative development.** (A-E) Protein enrichment calculated via mass spectrometry. Fold change represents the ratio of proteins purified from experimental (*VAL1-HA; val1-2; FRI-Sf2*) to control (*val1-2; fri*) samples, a t-test *P*-value is represented on the y-axis. Red dotted line indicates a *P*-value < 0.05 (resulting from the -log2 transformation of the actual p-value; proteins above the line have a significant enrichment). Each bubble represents an individual protein, the size or the bubble represents the protein abundance averaged across the experimental and control samples. (A, C-E) Proteins immunoprecipitated using an anti-HA antibody, (B) total proteome samples.

samples (Fig 7B). Comparisons of the proteins present in samples harvested from 1, 2 and 3 weeks old plants revealed that ATBMI1A was consistently among the most abundant proteins immunoprecipitated with VAL1-HA (Fig 7C, 7D and 7E). The abundance of ATBMI1A in the immunoprecipitated sample increased significantly from 1 week to 2 weeks, as indicated by both the increase in the fold change between experimental and control samples, and the statistical significance of the enrichment. This is probably a result of the increase in the abundance of VAL1 between 1 and 2 weeks (Fig 6D and 6E), a result which was confirmed by the increasing abundance of VAL1 in the immunoprecipitated samples from different time points (Fig 7C, 7D and 7E). The parallel changes in the abundance of VAL1-HA and ATBMI1A, and the absence of any major change in the proteins associated with VAL1-HA in different samples, suggest that the binding partners of VAL1 do not change significantly during shoot development.

To investigate overall trends in protein accumulation during vegetative development we conducted a supervised clustering analysis of the *val1-2; VAL1-HA; FRI-Sf2* total proteome sample. We designed two clusters in which proteins either increased or decreased from 1 to 2 to 3 weeks of development (see Methods for details). The 50 proteins with the strongest and most consistent decreasing developmental trend were significantly enriched for Gene Ontology terms related to photosynthesis and carbon fixation (S7 Fig). In contrast, the 50 proteins with the highest increasing trend score were enriched for GO terms relating to water stress and translation. The finding that younger plants invest more resources in photosynthesis is consistent with a transcriptomic analysis of vegetative development in maize [51], and the enhanced photosynthetic capacity of juvenile plants at low light levels [3].

## Discussion

Plant life cycles are characterized by transitions between distinct developmental phases. *VAL* genes have previously been shown to promote the switch from embryogenesis to seed maturation [16,25], and from vegetative to reproductive growth [30,32]. The results presented here demonstrate that *VAL1* and *VAL2* also regulate the intervening transition from juvenile to adult stages of vegetative growth. *VAL* genes thus function as a regulatory hub that coordinates developmental transitions throughout plant life cycles.

### Regulation of miR156 expression by *VAL1/2*

Vegetative phase change is promoted by a temporal decline in miR156/miR157 expression [8]. When the level of miR156/miR157 falls below a specific threshold, the de-repression of *SPL* genes initiates a switch to adult identity. Previous work has shown that–with the exception of *SPL3* –the increase in *SPL* transcript levels during development is entirely attributable to post-transcriptional regulation by miR156/miR157 [11]. Factors that control the timing of vegetative phase change can therefore act in three ways: 1) by modifying the rate of decline in miR156; 2) by constitutively increasing or decreasing the level of miR156; and 3) by constitutively increasing or decreasing the rate of transcription of *SPL* genes. Our results suggest that *VAL1* and *VAL2* regulate vegetative phase change both by constitutively decreasing the level of miR156 and by indirectly promoting *SPL* gene expression.

Evidence that *VAL1/VAL2* constitutively regulate the level of miR156 was provided by the phenotype of plants deficient for *VAL1* and *VAL2*, VAL1-DNA binding patterns and from the phenotype of plants expressing a *MIR156C* transgene lacking VAL-binding sites. We found that although *VAL1* expression increases in the shoot apex as plants develop, *val1; val2* double mutants displayed only a slight decrease in the rate at which miR156 declines. Instead, *val1; val2* double mutants exhibited a significant increase in the level of miR156 at every stage of vegetative development we examined. Consistent with this result, *mir156/mir157* mutants transformed with a *MIR156C* transgene lacking VAL-binding sites had elevated levels of miR156 relative to the wild type *MIR156C* control, but displayed the same temporal decrease in miR156 as control plants.

We have proposed that the decrease in *MIR156A/C* expression during shoot development may be attributable to the stochastic replacement of H3K27ac and H3K4me3 by H3K27me3 [19,21]. The observation that the increase in miR156 expression in *val1; val2* is associated with a decrease in the level of H3K27me3 at *MIR156C* supports this hypothesis, in that it shows that H3K27me3 is associated with low levels of *MIR156C* expression.

The evidence that *VAL1/VAL2* regulate the level, but not the temporal expression pattern, of miR156 leaves open the question of how its temporal pattern arises. Notably, recent genome-wide studies have found that only ~45% of PRC2 targets are dependent on

recruitment by VALs/RY-motifs, and that the trans/cis-regulatory modules are unknown for ~36% of PRC2 targets [35,52]. Several other chromatin regulators, such as the CHD3 nucleosome remodeler PICKLE [19], the PRC-accessory protein LHP1 [53], the SWI/SNF2 chromatin remodeler BRAHMA [54], and the histone 2 regulators ARP6 and HTA9/11 [21,55] have been found to play a role in the expression of miR156. Furthermore, transcription factors including AGL15/18 [56], MYB33 [57], and members of the *NF-Y* family [58,59], also regulate miR156 expression. It is possible that the temporal expression pattern of miR156 is a consequence of complex interactions between these diverse factors, rather than being dependent on a single class of regulator, such as *VAL1/2*.

### *VAL* genes and PRC1 activity

*VAL1/2* are thought to be necessary for the recruitment of PRC1 to target loci, where it represses gene expression via PRC2-dependent and independent mechanisms [35,39,40]. We found that *VAL1/2* accelerate vegetative phase change by repressing the expression of *MIR156A/C*, and by indirectly promoting the expression of *SPL9*, a target of miR156. These results are consistent with a previous study [20], which showed that *VAL1/2* and the PRC1 components, AtBMI1A/B, repress *MIR156A/C* expression. However, the PRC1 components EMBRYONIC FLOWER1 [20] and RING1A/B [49] have also been reported to repress the expression of *SPL9* independently of miR156. These latter effects delay vegetative phase change, which is the exact opposite of the effect produced by PRC1-mediated repression of *MIR156A/C*. Together, these results indicate that PRC1 can operate at different points within a regulatory pathway, or in interacting regulatory pathways, to modulate the output of the pathway or pathways. If the genes repressed by PRC1 have different functions—as in the case of miR156 and its *SPL* targets—then the functional significance of a particular level of PRC1 activity at a particular locus can be difficult to predict. These results also support the hypothesis that there may be different forms of PRC1, which target different genes. Moreover, our finding that H2AK121ub deposition at *SPL9* was unaffected in *val1-5(sd); val2-3* suggests that VAL1/2 may not be universally required for PRC1-recruitment.

In this regard, it is interesting that although *MIR156A* and *MIR156C* are close paralogs and have similar expression patterns, previous studies [19,21], and the results presented here, indicate that these genes are differentially sensitive to mutations that affect the activity of PRC2 and PRC1. For example, we found that *val1; val2* mutants display a greater reduction in H3K27me3 at *MIR156C* than at *MIR156A*. Our findings align with the results of ChIP-seq studies of H2AK121ub and H3K27me3 in *atbmi1a/b/c* [40] and *val1-2; val2-3* [35] mutants. The implication of these observations is that *MIR156C* expression is more dependent on PRC1 and PRC2 activity than *MIR156A*. Defining the molecular basis for this difference could provide important insights into factors that influence epigenetic regulation in plants.

In addition to their roles in vegetative phase change, *VAL* genes regulate multiple nodes of the flowering time [29,30,32] and seed development [16,25–27] networks. The co-option of *VAL* activity throughout genetic networks thus appears critical to coordinating plant developmental transitions. Despite the centrality of *VAL* function to the control of developmental timing, the persistent and robust pattern of *MIR156A/C* expression in *val* mutant plants emphasizes the complexity of temporal regulation in plants.

## Materials and methods

### Plant material and growth conditions

All stocks were grown in the Col-0 background. The following genetic lines have been described previously: *val1-2* (SALK_088606), *val2-1* (CS906036) [25] (*val2-1* was backcrossed

to Col-0 6 times from the original Wassilewskija parent); *val2-3* (SALK_059568C) [16]; *clf-29* (SALK_021003) [60]; *swn-3* (SALK_050195) [17]; *mir156a-2 mir156c-1 mir157a-1 mir157c-1* [11]; *SPL9::rSPL9-GUS* [10]; *val1-2 VAL1::VAL1-3xHA FRI-Sf2* [30]. *val2-4* (SALK_127961) was obtained from the Arabidopsis Biological Resources Center (Ohio State University, OH, USA). Seeds were sown on fertilized Farfard #2 soil (Farfard) and kept at 4˚C for 3 days prior to transfer to a growth chamber, with the transfer day counted as day 0 for plant age (0 DAG–days after germination). Plant were grown at 22˚C under a mix of both white (USHIO F32T8/741) and red-enriched (Interlectric F32/T8/WS Gro-Lite) fluorescent bulbs in either long day (16 hrs light/8 hrs. dark; 40 µmol m$^{-2}$ s$^{-1}$) or short day (10 hrs light/14 hrs dark; 100 µmol m$^{-2}$ s$^{-1}$) conditions. The locus identifiers of the genes investigated in this study are as follows: *VAL1*, AT2G30470; *VAL2*, AT4G32010; *MIR156A*, AT2G25095; *MIR156C*, AT4G312877; *MIR157A*, AT1G66783; *MIR157C*, AT3G18217; *CLF*, AT2G23380; *SWN*, AT4G02020; *SPL9*, AT2G42200; *FLC*, AT5G10140. The *val1-5(sd)*, *val1-5(sd)*; *val2-3, val1-2*; *val2-4, val1-2*; *mir156ac mir157ac* and *VAL1::VAL1-GUS-VAL1 3'UTR* lines described herein have been donated to the ABRC (https://abrc.osu.edu/ stock numbers CS72451-CS72455 respectively).

## Identification of the *val1-5(sd)* mutant

The *val1-5(sd)* allele was generated by exposing *mir157a-1; mir157c-1* seed to ethyl methane-sulfonate. An M2 mutant plant exhibiting delayed vegetative phase change was backcrossed to the parental line and allowed to self. Tissue was pooled from 30 plants exhibiting severely delayed vegetative phase change in the BC1F2 generation. DNA was extracted via-SDS lysis and phenol-chloroform extraction and further purified using Clean and Concentrator columns (Zymo Research). DNA concentration was determined using a Qubit 2.0 Fluorometer (Invitrogen) and 1µg of DNA sheared using a Covaris S2 sonicator (Covaris) to produce 350bp inserts. Sequencing libraries were made following the TruSeq DNA PCR-free LT Sample Prep Kit (Illumina) manufacturer's instructions. Library quality and quantity was validated by Bioanalyser (Agilent) and KAPA analysis (Kapa Biosystems). 100bp paired end reads were generated using a HiSeq 2500 (Illumina) and aligned to the TAIR10 reference genome following the default SHORE pipeline [61]. The SHOREmap backcross pipeline [62] using default options was employed to identify polymorphisms. Manual inspection of allele frequencies in the mutant revealed a peak centered on the *VAL1* locus. The causative mutation was confirmed by Sanger sequencing and complementation assays. The mutant was backcrossed to a 'Traffic Line' [63] with seed-fluorescent markers inserted adjacent to the *VAL1* locus to eliminate additional closely linked polymorphisms. Consequently, the resultant *val1-5(sd)* plants used in this study contain a linked *pNAP::RFP* insertion at 13,622,737bp on Chromosome 2 (Crick strand).

## Generation of transgenic plants

For RY-mutation lines the RY-site TGCATG was replaced by TGTATG. A 5kb *MIR156C* genomic sequence including 2kb upstream of the transcriptional start site and 665bp downstream of the end of the last exon was cloned into the binary vector pAGM4723 from the Golden Gate MoClo toolbox supplied by Addgene (www.addgene.org) [64,65]. For -RY3, -RY4 and -RY5 Q5 Site Directed Mutagenesis Kit (New England Biolabs) was used to induce a substitution directly into the expression vector. For -RY1, -RY2 and -RY12345 Gibson Assembly cloning (New England Biolabs) was used to assemble individual fragments into the same backbone. Golden Gate cloning was also used to generate *VAL1::VAL1-VAL1 3'UTR*, *VAL1::VAL1-GUS-VAL1 3'UTR*, *VAL1::GUS-VAL1 3'UTR* lines. Promoter/5'UTR (*VAL1*–2.3kb), functional (*VAL1*–3.8kb, *MIR156A* [7]) and 3'UTR/terminator (*VAL1* – 2kb) sequences were

cloned separately from *Arabidopsis* gDNA, with Type II restriction sites removed where necessary. *GUS* and *AtuOCS* sequences were obtained from the MoClo Plant Parts toolkit supplied by Addgene (www.addgene.org) [66]. Component parts were assembled using Golden Gate cloning into the pAGM4723 binary vector, including green or red seed fluorescent expression cassettes as selectable markers. Constructs were transformed into *Arabidopsis* using the floral dip method. All primers used for cloning are included in S1 Table.

## Quantification of gene expression

Tissue (either shoot apices with leaf primordia ≤1mm attached or isolated leaf primordia 0.5-1mm in size–as specified in the text) were ground in liquid nitrogen and total RNA extracted using Trizol (Invitrogen) as per the manufacturer's instructions. RNA was treated with RNAse-free DNAse (Qiagen) and 250ng-1μg of RNA was used for reverse transcription using Superscript III (Invitrogen). Gene specific stem-loop RT primers were used to amplify miR156, miR157, miR172 and SnoR101 sRNAs [67,68] and a polyT RT primer was used for mRNA amplification. Three-step qPCR of cDNA was carried out using SYBR-Green Master Mix (Bimake). qPCR reactions were run in triplicate and an average was calculated. Relative transcript levels were normalized to snoR101 (for miRNAs) and *ACT2* (for mRNAs) and expressed as a ratio of expression to a specified control sample. The qPCR primers used in this study are listed in S1 Table.

## Chromatin immunoprecipitation

Expanded leaves and roots were removed during tissue harvesting to produce samples enriched for shoot apices and young leaves. For histone ChIP ~0.5g of fresh tissue per antibody and for anti-HA ChIP ~5g of fresh tissue were harvested. Samples were fixed in 1% formaldehyde under vacuum for 15 minutes. Cross-linked samples were ground in liquid nitrogen and suspended in Honda buffer (0.44M sucrose, 1.25% ficoll, 2.5% dextran 40, 20mM hepes pH 7.4, 10mM MgCl$_2$, 0.5% Triton, 5mM DTT, 1mM PMSF, 1% protease inhibitors), filtered through two layers of Miracloth (EMD Millipore), and pelleted and washed thrice in Honda buffer. For histone ChIP, pellets were resuspended in nuclei lysis buffer (50mM Tris-HCl pH 8, 10mM EDTA, 1% SDS, 1% protease inhibitors), for anti-HA ChIP, pellets were resuspended in RIPA buffer (1X PBS, 1% NP-40, 0.5% sodium deoxycholate, 0.1% SDS, 1% protease inhibitors). Samples were sonicated using a Fisherbrand Sonic Dismembrator (Fisher Scientific) 6x 10s at setting 3.2. ChIP samples were pre-cleared using Dynabeads Protein A (Invitrogen). 2% was removed as input and samples were incubated overnight with 1% antibody (for histone ChIP: anti-H3 (abcam ab1791, RRID:AB_302613), anti-H3K27me3 (EMD Millipore 07–449, RRID:AB_310624), anti-H2AK121ub (Cell Signaling Technology 8240, RRID:AB_10891618); for VAL1-HA ChIP: anti-HA (Roche 11583816001, RRID:AB_514505)). Chromatin-antibody conjugates were purified with Dynabeads Protein A and washed in low/high salt, lithium and TE buffers. Following reverse-crosslinking DNA was isolated using a QIAquick PCR Purification Kit (Qiagen).

For ChIP-qPCR assays, three-step qPCR was carried out using SYBR-Green Master Mix (Bimake). qPCR reactions were run in triplicate and an average was calculated. Data were normalized and presented as follows: 1) For H3K27me3 –*STM* was used as a control locus [22,69,70], data is presented as a ratio of (H3K27me3 gene of interest/H3 gene of interest) to (H3K27me3 *STM*/H3 *STM*); 2) For H2AK121ub–*ABI3* was used as a control locus [16,30,49], data is presented as a ratio of (H2AK121ub gene of interest/input gene of interest) to (H2AK121ub *ABI3*/input *ABI3*); 3) For VAL1-HA–*UBQ10*, *ACT7* and *TA3* were used as control loci, data is presented as a ratio of ((VAL1-HA ChIP gene of interest/input gene of

interest)/(VAL1-HA ChIP control/input control)) relative to ((WT ChIP gene of interest/
input gene of interest)/(WT ChIP control/input control)). The qPCR primers used in this
study are listed in S1 Table.

## Mass spectrometry

*VAL1::VAL1-3xHA; val1-2; FRI-Sf2* and *val1-2* genotypes were used as experimental and
control samples respectively. Expanded leaves and roots were removed during tissue harvest-
ing to produce samples enriched for shoot apices and young leaves. 2-3g of fresh tissue was
harvested for immunoprecipitation, and 0.2–0.6g was harvested for total protein extraction,
at 1,2 and 3 weeks after germination. For immunoprecipitation, tissue was ground in liquid
nitrogen, suspended in IP buffer (20mM Tris-HCl pH 8, 150mM NaCl, 2.5mM EDTA, 0.5%
Triton, 1% protease inhibitors, 1mM PMSF), rotated for 2 hours at 4˚C and filtered through
2 layers of Miracloth. Anti-HA (Roche) conjugated Dynabeads Protein A (Invitrogen) were
added and samples were rotated overnight at 4˚C. Beads were washed thrice with IP buffer
and proteins were purified for mass spectrometry using an S-Trap: Rapid Universal MS Sam-
ple Prep (Protifi) following the manufacturer's instructions. For total proteomes, ground tis-
sue was suspended in 8M urea and rotated at room temperature for 45 minutes. The samples
were centrifuged thrice and the supernatant reduced with DTT (final concentration 5mM)
and alkylated with iodoacetamide 40 (final concentration 40mM) before overnight digest
with trypsin. Samples were resuspended in 10 μl of water + 0.1% TFA and loaded onto a Dio-
nex RSLC Ultimate 300 (Thermo Scientific, San Jose, CA, USA), coupled online with an
Orbitrap Fusion Lumos (Thermo Scientific). Chromatographic separation was performed
with a two-column system, consisting of a $C_{18}$ trap cartridge (300 μm ID, 5 mm length) and
a picofrit analytical column (75 μm ID, 25 cm length) packed in-house with reversed-phase
Repro-Sil Pur $C_{18}$-AQ 3 μm resin. Peptides were separated using a 90 min gradient (for the
IP experiments) and 180 min (for the full proteome experiment) from 2–28% buffer-B
(buffer-A: 0.1% formic acid, buffer-B: 80% acetonitrile + 0.1% formic acid) at a flow rate of
300 nl/min. The mass spectrometer was set to acquire spectra in a data-dependent acquisi-
tion (DDA) mode. Briefly, the full MS scan was set to 300–1200 m/z in the orbitrap with a
resolution of 120,000 (at 200 m/z) and an AGC target of 5x10e5. MS/MS was performed in
the ion trap using the top speed mode (2 secs), an AGC target of 10e4 and an HCD collision
energy of 30. Raw files were searched using Proteome Discoverer software (v2.4, Thermo Sci-
entific) using SEQUEST as search engine using the SwissProt *Arabidopsis thaliana* database.
The search for total proteome included variable modifications of methionine oxidation and
N-terminal acetylation, and fixed modification of carbamidomethyl cysteine. Trypsin was
specified as the digestive enzyme. Mass tolerance was set to 10 pm for precursor ions and 0.2
Da for product ions. Peptide and protein false discovery rate was set to 1%. Data transforma-
tion, normalization and statistical analysis using heteroscedastic t-test was performed as pre-
viously described [71].

Proteins were sorted according to their descending or ascending linearity across the three
week experimental time course. To do so, we used a custom score taking into account mono-
tonic trend, reproducibility across replicates and magnitude of change across weeks. The
50 proteins with the highest descending and ascending trend scores were used to identify
enriched GO terms for biological processes using the Fisher's Exact PANTHER Overrepresen-
tation Test (released 2020-07-28) and GO Ontology database DOI: 10.5281/zenodo.4081749
(released 2020-10-09). The *Arabidopsis thaliana* genome was used as a reference list. Protein
interaction maps for the same sets of 50 proteins were made using the STRING app from
Cytoscape [72].

## GUS staining and histology

Shoot apices and whole plants were fixed in 90% acetone on ice for 10 minutes, washed with GUS staining buffer (5mM potassium ferricyanide and 5mM ferrocyanide in 0.1M $PO_4$ buffer) and incubated at 37˚C overnight in GUS staining buffer with 2mM X-Gluc. Embryos were placed directly in X-Gluc (GoldBio) GUS staining buffer and incubated for 1hr. To quantify GUS activity, a 4-methylumbelliferyl b-D-glucuronide (MUG) (Sigma-Aldrich) assay was carried out as previously described [11]. For histological observations individuals were fixed in FAA (3.7% formaldehyde), dehydrated in an ethanol series and cleared using Histo-Clear (National Diagnostics). Following embedding in Paraplast Plus (Sigma-Aldrich) 8μM sections were produced using an HM 355 microtome (Microm) and visualized using an Olympus BX51 microscope with a DP71 camera attachment (Olympus).

## Leaf measurements

The angle of the leaf blade base was measured using two tangents from the blade base intersecting at the petiole. Blade length:width ratios were measured using the tip of the blade to the petiole junction (length) and the widest point of the blade. Measurements of leaf shape were made using ImageJ [73].

## Quantification and statistical analyses

Details of all statistical analyses, including the type of statistical test, sample size, replicate number and significance threshold, are included in the relevant Fig legend. For figures featuring boxplots, boxes display the IQR (boxes), median (lines), and values beyond $1.5^*$ IQR (whiskers); mean values are marked by a solid diamond (♦). Statistical analyses were carried out using RStudio [74] and Microsoft Excel.

## Supporting information

**S1 Fig. *val1-5(sd)* is an antimorphic allele.** (A) Sequence alignment of the B3 DNA-binding domain N-arm of *Arabidopsis* LAV family members and the maize ABI3 ortholog VP1. Numbers correspond to amino acid sequence position, colors correspond to the ClustalX amino acid color scheme. In the *val1-5(sd)* mutant a C>T base substitution converts an arginine to a cysteine. (B) *val1-5(sd)* complementation test with the null *val1-2* allele. (C) Rescue of the *val1-5(sd)* abaxial trichome phenotype with a *VAL1* genomic sequence. Independent T1 lines are shown. (D) Allele heterozygosity testing. (B-D) Boxes display the interquartile range (IQR) (boxes), median (lines) and values beyond $1.5^*$IQR (whiskers); mean values are marked by υ. Samples sizes are displayed on the graph. Statistically distinct genotypes were identified by one-way ANOVA with *post hoc* Tukey multiple comparison test (letters indicate statistically distinct groups; $P < 0.05$), all plants grown in LD. (E) qRT-PCR analysis of gene expression in whole seedlings harvested at 7 DAG in LD conditions. Each data point represents a biological replicate and is the average of three technical replicates. Bars represent the mean and error bars mean±s.e.m. Asterisks represent significant difference between WT and *val2-4* calculated by an unpaired two-tailed *t*-test (*** $P < 0.0005$). (F) Heteroblastic series of lines shown in Fig 1.
(TIF)

**S2 Fig. *VAL* genes redundantly regulate miR156 expression.** qRT-PCR analyses of gene expression in shoot apices with leaf primordia (LP) ≥ 1mm removed at 1, 2 and 3 weeks. All plants were grown in SD conditions. Each data point represents a biological replicate and is the average of three technical replicates. Coloured lines represent the mean and black lines

mean±s.e.m. Asterisks represent significant differences between WT and *val* mutants at the same time point, calculated by an unpaired two-tailed *t*-test with a Bonferroni correction for multiple comparisons (* $P < 0.025$; ** $P < 0.005$; *** $P < 0.0005$).
(TIF)

**S3 Fig. VAL1 binds consistently to *MIR156A* and *MIR156C* during vegetative development.** (A) Schematic depicting the location of primers used for ChIP-qPCR, the sequences encoding the miR156 hairpin and mature miRNA are coloured green and blue respectively. (B) Anti-HA ChIP-qPCR of WT Col control plants at 2W and *VAL1::VAL1-HA; val1-2; FRI-Sf2* plants at 1, 2 and 3W of growth. The data is presented as percentage input normalized to a control locus (*UBQ10* or *TA3*) and is displayed relative to WT. Each data point represents a biological replicate and is the average of three technical replicates, bars represent the mean and error bars the mean±s.e.m. Plants were grown in SD conditions.
(TIF)

**S4 Fig. RY sites are not required for the temporal decline of *MIR156C*.** (A) qRT-PCR analyses of gene expression in shoot apices with LP $\geq$ 1mm removed at 1 and 2 weeks. Bars represent the average of three technical replicates for a single biological replicate of pooled T1 plants, at least 15 independent T1 plants were pooled for each sample. *qm = mir156a mir156c mir157a mir157c quadruple mutant*, *tm = mir156a mir157a mir157c triple mutant*. Plants were grown in SD conditions.
(TIF)

**S5 Fig. *VAL* genes do not regulate *SPL9* chromatin state.** (A, B) Temporal analysis of histone modification scalculated by ChIP-qPCR. Each data point represents a biological replicate and is the average of three technical replicates. Lines represent the mean and bars represent the mean±s.e.m., (A) H2AK121ub values are relative to input and normalised to *ABI3* as an internal control. (B) H3K27me3 values are relative to H3 and normalised to *STM* as an internal control. Plants were grown in SD conditions.
(TIF)

**S6 Fig. *VAL1* expression is dependent on genetic elements in the coding sequence of the gene.** (A) Torpedo-stage embryos of two-independent homozygous transgenic lines each expressing a transcriptional (top panel) or translational (bottom panel) *VAL1-GUS* reporter construct. (B) Seedlings at 3 DAG. Each number designates an independent homozygous transgenic line. Arrow heads point to initiating lateral root primordia. All plants were grown in LD.
(TIF)

**S7 Fig. Proteomic changes during vegetative development.** (A,B) The 50 proteins with the strongest increasing or decreasing trend score during vegetative development in the experimental sample only (see Methods for details). (A) GO terms enriched within the 50 proteins that have the strongest increasing or decreasing trend score. (B) Interaction networks for each set of 50 proteins, the darker the color the stronger the increasing (red) or decreasing (blue) trend score.
(TIF)

**S1 Table. Primer sequences.** This table includes all the primer sequences used in this study.
(XLSX)

**S1 Data. Proteins detected by mass spectrometry following anti-HA immunoprecipitation.** Related to Fig 7. This dataset includes the raw mass spectrometry results and data processing

for IP-samples.
(XLSX)

**S2 Data. Total proteome changes during vegetative development.** Related to Fig 7. This dataset includes the raw mass spectrometry results and data processing for total proteome samples.
(XLSX)

**S3 Data. Underlying data for all figures.** This dataset includes all the data used to generate Figs 1–6 and S1–S6 Figs.
(XLSX)

# Acknowledgments

We thank members of the R.S.P laboratory and Doris Wagner (University of Pennsylvania) for helpful discussions, Daniel Martinez and Ryan McCarthy (University of Pennsylvania) for assistance with histology and chromatin sonication, the Arabidopsis Biological Resource Center for T-DNA insertion lines and Julia Qüesta and Caroline Dean (John Innes Center) for the kind donation of *VAL1-HA; val1-2; FRI-Sf2* seed.

# Author Contributions

**Conceptualization:** Jim P. Fouracre, R. Scott Poethig.

**Formal analysis:** Jim P. Fouracre, Simone Sidoli.

**Funding acquisition:** R. Scott Poethig.

**Investigation:** Jim P. Fouracre, Jia He, Victoria J. Chen, Simone Sidoli.

**Methodology:** Jim P. Fouracre, R. Scott Poethig.

**Project administration:** R. Scott Poethig.

**Resources:** Jim P. Fouracre, Jia He.

**Supervision:** R. Scott Poethig.

**Visualization:** Jim P. Fouracre, Simone Sidoli.

**Writing – original draft:** Jim P. Fouracre.

**Writing – review & editing:** Jim P. Fouracre, Jia He, Simone Sidoli, R. Scott Poethig.

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
