## [Decision Letter · Decision Letter 0]

21 Apr 2021

Dear Dr Fouracre,

Thank you very much for submitting your Research Article entitled 'VAL genes regulate vegetative phase change via miR156-dependent and independent mechanisms' to PLOS Genetics.

The manuscript was fully evaluated at the editorial level and by independent peer reviewers. The reviewers appreciated the attention to an important topic but identified some concerns that we ask you address in a revised manuscript

We therefore ask you to modify the manuscript according to the review recommendations. Your revisions should address the specific points made by each reviewer.

[LINK]

Yours sincerely,

Claudia Köhler

Section Editor: Plant Genetics

PLOS Genetics

Reviewer's Responses to Questions

**Comments to the Authors:**

Reviewer #1: The paper provides a very detailed and comprehensive analysis of the role of the transcription factor VAL1 in regulating phase change in Arabidopsis. Overall I found it an impressive and high quality study, with elegant genetic analysis, so very suitable for PLoS Genetics.

The paper provides solid evidence that VAL1 and the related VAL2 gene promote phase change so that double mutants in particular have a prolonged juvenile phase. This is correlated with increases in miR156 levels, binding of VAL1 to RY elements in miR156 genes, and they also show that mutating the RY elements results in increased miR156 expression levels. However, val1 mutations continue to have an effect on phase change in quadruple mutants lacking most miR156 and miR157 gene activity, or in such mutants carrying miR156 transgenes lacking RY sites. Thus, although there is a good correlation of the effects of val1 mutations on phase change with increased miR156 expression, VAL1 can also act independently of miR156 and it is not very clear how much of the effects on phase change are due to the increases in miR156 activity. Consistent with this, they also show that VAL1 can regulate SPL9 activity independently of miR156/7, and that this is most likely indirect as VAL1 (a repressor) promotes SPL9 activity.

The paper also looks via ChIP at the effects of VAL1/2 on H3K27me3 and H2AK121ub levels at miR156 levels, with H3K27me3 levels increasing with time, and the rate of increase slowing in val1 mutants especially at miR156c. There seem to be little effect on val1 val2 double mutants on levels early in development (week one) although miR156c expression is clearly elevated at week one in the double mutants. Genetic analysis indicates that val1 mutation acts synergistically with clf or swn mutations affecting PRC2 (the H3K27me3 histone methyltransferase), this is interpreted as the genes acting in the same pathway which is plausible given the various redundancies involved but not straightforward, and is not well correlated with increases in miR156/7 levels, so may act via alternative pathways. There is also a decrease in H2AK119ub at miR156c in particular in val1 2 doubles, which may contribute to the elevated expression.

Another clear finding of the paper is that VAL1/2 affect overall expression levels of miR156 but not the dynamics, i.e. there is still a marked decrease in expression with time, suggesting there are other key regulators determining this drop. Reporter gene analysis suggests that VAL1 expression increases with time, perhaps contributing to increased H3K27me3 and repression with age. It might have been interesting to test this further, e.g. by VAL1 over/mis expression, but the paper already contains a very large amount of data.

In conclusion I think this is an important paper, well done and pretty honest in pointing out the things that don’t fit with a straightforward model. The picture that emerges is slightly cloudy, i.e VAL genes promote adult phase transition, in part by regulating epigenetic changes at miR156 and overall expression levels, but they are also acting independently via other less well defined pathways, and the relative significance of the two is not wholly clear. Nonetheless I find it a very thorough study and extremely suitable for PLoS Genetics.

Very minor comments – Materials and methods did not explain at well how miR156 and miR157 mature transcript levels were quantified. Probably this was by very standard methods, but they should be described in better detail. Lines 572 – 574 states that VAL1 VAL 2 repress SPL9 expression by miR156 independent means, however the results section showed the opposite e.g. expression of miR156 resistant SPL9 transgene is much higher in VAL1+ than in val1 mutants so VAL1 activity promotes SPL9 expression albeit likely indirectly.

Reviewer #2: In addition to the known mechanism of juvenile-to-adult phase transition that relies on the VAL-mediated PRC recruitment and repression of miR156 and miR157, the authors describe a novel finding that VAL TFs act in a mode that involves transcriptional regulation of the SPL genes but is independent of miR156. VAL TFs are shown to moderate the transcription level of the miR156 miR157 genes but their temporal decline during development is shown to be VAL-independent. Based on a VAL1-HA coIP and MS experiment, the authors identify the AtBMI1A as a robust VAL1 interactor, confirming previous studies, and suggest that the spectrum of VAL1 interactors is relatively stable during the first 3 weeks of shoot development when the juvenile-to-adult phase transition takes place.

The manuscript is clearly written, methods are well-described, the experiments are carefully conducted and interpreted, and the data presented supports the conclusions. Even though the exact mechanism of how the VAL TFs contribute to the miR156-independent mode of action is not uncovered, the manuscript opens and starts filling a gap in our understanding of the VAL-mediated juvenile-to-adult transition and the role of the VAL transcription factors in general.

I have several comments:

1. Line 157 – val2-4 is a newly described val2 allele – has expression of the locus bee tested to determine the effect of the T-DNA insertion?

2. Fig 1E (line 163), Fig.4D – The blade base angle results are not intuitive. For WT, the expectation would be an increase of blade base angle in adult leaves. In contrast, the angle decreases in leaf 5 compared to leaf 3. Can the authors indicate how the blade angle is defined in this particular quantification?

3. Fig1C – the color-coding should be changed as the shades of blue and green are difficult to be differentiated in pdf or print

4. Lines 164-166: A comparison is made between val1-5(sd); val 2-3 and val1-2;val2-3, where the latter is not shown (analysed) and a reference to previous works (in which the mutants are grown in vitro on MS medium) is made. I would argue that the difference may be caused by different growth conditions and if such comparison should be made, the mutants should be tested in parallel. I therefore think that the association between the val1-5(sd) semi-dominance and VAL2 interaction is not sufficiently justified.

5. Lines 203-204 (Fig 2A,B, S2 Fig) While I agree with the increase of miR156 in val mutants, I don´t think evidence supports the general conclusion that miR157 is increased upon loss of VAL activity. Increase is robustly visible in val1-2; val2-1 (Fig S2) but not in the other mutant/allelic combinations. In fact, there is significantly less miR157 in val1-5(sd); val 2-3 in youngest leaf primordia.

6. Line 229: formulation: “…we examined PRC2 activity in the val1-5(sd); val2-3…” PRC2 activity was not examined (and most likely it would be comparable if “only” recruitment is impaired). Please reformulate.

7. ChIP-qPCR – the rationale behind the difference in quantification is unclear – Fig3B – H3K27me3 related to H3 and normalized to STM (H3K27me3 target) and Fig. 3F – H2Aub related to input and normalized to ACT7 (H3K27me3/H2Aub non-target). The stability of STM and ACT7 in terms of the modification occupancy is not shown.

Line764, line 510 – 512: VAL1-HA coIP experiment- the age of plants at harvest should be indicated in methods as well as in the main text/legend.

**Have all data underlying the figures and results presented in the manuscript been provided?**

Reviewer #1: Yes

Reviewer #2: Yes

PLOS authors have the option to publish the peer review history of their article (what does this mean?). If published, this will include your full peer review and any attached files.

Reviewer #1: No

Reviewer #2: No

---

## [Decision Letter · Decision Letter 1]

28 May 2021

Dear Dr Fouracre,

We are pleased to inform you that your manuscript entitled "VAL genes regulate vegetative phase change via miR156-dependent and independent mechanisms" has been editorially accepted for publication in PLOS Genetics. Congratulations!

Yours sincerely,

Claudia Köhler

Section Editor: Plant Genetics

PLOS Genetics

Claudia Köhler

Section Editor: Plant Genetics

PLOS Genetics

Comments from the reviewers (if applicable):

Reviewer's Responses to Questions

**Comments to the Authors:**

Reviewer #1: authors have addressed my comments.

Reviewer #2: The revised version of the manuscript addresses all the comments raised by the reviewers and led in mostly to minor modifications of the manuscript.

One exception to this is changing the quantification strategy in an anti-H2Aub ChIP experiment, which changed the conclusions of that experiment (Fig 3F, results lines 254 - 258, discussion lines 517-523 of changes-tracked manuscript), newly suggesting limited effect of the VAL TFs on the temporal deposition of H2Aub at MIR156A and MIR156C genes. This demonstrates the fragile nature of ChIP quantification strategies, especially in cases of limited effect/trend. The conclusions on H3K27me3 deposition however seem to be robust, having been observed repeatedly and with a clear trend. Although the particular conclusion of the one experiment has been altered, this does not change the main findings and conclusions of the manuscript.

All the other reviewer comments have in my opinion been sufficiently and carefully addressed.

**Have all data underlying the figures and results presented in the manuscript been provided?**

Reviewer #1: Yes

Reviewer #2: Yes

PLOS authors have the option to publish the peer review history of their article (what does this mean?). If published, this will include your full peer review and any attached files.

Reviewer #1: No

Reviewer #2: No

**Data Deposition**

http://datadryad.org/submit?journalID=pgenetics&manu=PGENETICS-D-21-00390R1

**Press Queries**

---

## [Editor Report · Acceptance letter]

22 Jun 2021

PGENETICS-D-21-00390R1 

*VAL* genes regulate vegetative phase change via miR156-dependent and independent mechanisms 

Dear Dr Fouracre, 

We are pleased to inform you that your manuscript entitled "*VAL* genes regulate vegetative phase change via miR156-dependent and independent mechanisms" has been formally accepted for publication in PLOS Genetics! Your manuscript is now with our production department and you will be notified of the publication date in due course.

With kind regards,

Zita Barta

PLOS Genetics

On behalf of:
